# Adipose Tissue-Derived Mesenchymal Stem Cells Extend the Lifespan and Enhance Liver Function in Hepatocyte Organoids

**DOI:** 10.3390/ijms242015429

**Published:** 2023-10-21

**Authors:** Sun A Ock, Seo-Yeon Kim, Won Seok Ju, Young-Im Kim, Ha-Yeon Wi, Poongyeon Lee

**Affiliations:** Animal Biotechnology Division, National Institute of Animal Science, Rural Development Administration, 1500 Kongjwipatjwi-ro, Iseo-myeon, Wanju-gun 55365, Republic of Korea

**Keywords:** porcine, hepatocyte, organoid, liver, adipose-derived mesenchymal stem cells

## Abstract

In this study, we generated hepatocyte organoids (HOs) using frozen-thawed primary hepatocytes (PHs) within a three-dimensional (3D) Matrigel dome culture in a porcine model. Previously studied hepatocyte organoid analogs, spheroids, or hepatocyte aggregates created using PHs in 3D culture systems have limitations in their in vitro lifespans. By co-culturing adipose tissue-derived mesenchymal stem cells (A-MSCs) with HOs within a 3D Matrigel dome culture, we achieved a 3.5-fold increase in the in vitro lifespan and enhanced liver function compared to a conventional two-dimensional (2D) monolayer culture, i.e., more than twice that of the HO group cultured alone, reaching up to 126 d. Although PHs were used to generate HOs, we identified markers associated with cholangiocyte organoids such as cytokeratin 19 and epithelial cellular adhesion molecule (*EPCAM*). Co-culturing A-MSCs with HOs increased the secretion of albumin and urea and glucose consumption compared to HOs cultured alone. After more than 100 d, we observed the upregulation of tumor protein *P53 (TP53)-P21* and downregulation of *EPCAM*, albumin (*ALB*), and cytochrome P450 family 3 subfamily A member 29 (*CYP3A29*). Therefore, HOs with function and longevity improved through co-culturing with A-MSCs can be used to create large-scale human hepatotoxicity testing models and precise livestock nutrition assessment tools.

## 1. Introduction

To address severe liver damage arising from diverse factors such as viral infections, obesity, and alcohol consumption, extensive research is underway, focusing on new drug development, cell therapy products, and bio-artificial liver devices [1]. To achieve these goals, it is essential to develop a cell source capable of maintaining liver-specific functions for a long period and an in vitro culture system optimized for this purpose.

Primary hepatocytes, obtained from humans and animals, are optimal choices for liver disease research, but have limited availability owing to supply constraints. In addition, unlike in vivo hepatocytes, they do not maintain liver or cell proliferation function under traditional monolayer culture conditions [2,3]. To overcome these limitations, research has been directed towards differentiating hepatocytes from pluripotent stem cells (PSCs) [3,4] known for their robust self-renewal capabilities. Alternatively, the induced dedifferentiation of somatic cells into hepatocyte-like cells has been explored by introducing liver-specific transcription factors [5,6]. Nevertheless, hepatocyte-like cells derived from PSCs often exhibit characteristics akin to those of early cholangiocytes or fetal hepatocytes, making them not directly comparable to fully mature hepatocytes. In addition, PSC-derived hepatocyte-like cells have the disadvantage of undergoing a long and multi-step differentiation induction process; therefore, they cannot be used immediately when needed.

Conventional in vitro cell culture methods such as monolayer two-dimensional (2D) culture have limitations in mimicking the functions of tissues and organs in vivo. To overcome these limitations, organoid culturing was performed using three-dimensional (3D) cultures of various epithelial cells from the intestine, liver (bile duct), and lungs (alveolar/airway) [7,8,9]. The organoids, when compared with 2D-cultured cells, have been reported to better mimic the characteristics of tissues and organs. However, the liver organoids reported in previous studies have limitations. They primarily comprise hollow bile duct-derived structures that often require maturation or have inadequate functional capabilities. In contrast, the liver parenchyma is composed of hepatocytes responsible for synthesizing crucial proteins such as serum proteins [9,10]. Studies have been conducted to develop hepatocyte organoids using fetal or adult hepatocytes from humans and mice and to proliferate them in vitro [10,11]. However, these organoids have lower in vitro prolife ration ability than bile duct-derived organoids.

Researchers have studied the effects of co-culturing hepatocytes or hepatocyte-like cells with various cell types, such as fibroblasts and mesenchymal stem cells (MSCs), on extracellular matrices [12,13,14,15]. While this approach can enhance the function and proliferative capacity of hepatocytes in vitro, the resulting cell structures resemble cell aggregates more than the hepatocyte organoids formed using the Matrigel-based 3D-dome organoid culture technique.

Pigs are used as models for human xenotransplantation and disease research because they are physiologically similar to humans and can be genetically modified [16]. However, concerns about animal rights have led to the development of alternative testing methods to reduce animal use in research. Most studies have focused on developing alternative testing methods for humans; however, we developed pig hepatocyte organoids as part of an alternative testing method for animals. We also succeeded in culturing functional hepatocyte organoids (HOs) for more than 100 d by co-culturing adipose tissue-derived MSCs (A-MSCs) with hepatocyte organoids.

## 2. Results

### 2.1. Generation of HO

The survival rate of freeze-thawed primary hepatocytes was significantly improved by using 36% Percoll treatment, as evidenced by both the 0.4% trypan blue exclusion test and the downregulation of apoptosis regulatory genes B-cell lymphoma 2 (*BCL2*) antagonist/killer 1 (*BAK1*) and cysteine aspartase 8 (*CASP8*) (Appendix A). Consequently, 36% Percoll-treated primary hepatocytes (PHs) were used as the control group in all subsequent experiments. To establish the optimal cell concentration for HO generation, 1000 (1000 group), 10,000 (10,000 group), 20,000 (20,000 group), and 50,000 (50,000 group) hepatocytes were seeded in each well. The number of HOs increased in a concentration-dependent manner until day 14 of in vitro culture (Figure 1A–C). The cell seeding concentration of 10,000 to 50,000 cells per well did not affect the diameters of the HOs, which averaged 160 to 180 um at 20 d. For further analysis, the major liver function gene albumin (*ALB*) and cell apoptosis genes *BAK1* (pro-apoptotic) and *BCL2* (anti-apoptotic) were examined in the groups seeded with 20,000 and 50,000 cells, respectively, in which large numbers of HOs were observed. In the 50,000 group, compared to the 20,000 group, the edges of the HOs were not smooth, and dark cell debris was observed from days 14 to 20. Although there was no change in *ALB* expression in either group, an increase in *BAK1* expression was observed in the 50,000 group (Figure 1E).

Owing to the limited cell culture space, a seeding concentration of 20,000 cells for 14 d was deemed appropriate for generating HOs, and subsequent experiments were conducted under these conditions.

### 2.2. Expression of Liver-Specific Genes in HOs

HOs were assessed using RT-qPCR experiments to analyze the expression of genes related to liver function, hepatic progenitor factors, and drug degradation compared to 2D cultures (Figure 2). For the experiment, PHs were employed as the positive control group while ear fibroblasts (EF) were used as the negative control group. Genes responsible for the synthesis of major liver proteins, including *ALB*, alpha-fetoprotein (*AFP*), transferrin (*TF*), and Alpha-1 antitrypsin (*AAT*), were upregulated in 3D cultures and HOs compared with those in 2D cultures (Figure 2A).

Among the bile duct/hepatic progenitor cell factors such as keratin 19 (*KRT19*), epithelial cellular adhesion molecule (*EPCAM*), and SRY-box transcription factor 9 (SOX9), *KRT19* and *SOX9* displayed significant reductions of 225.2-fold and 32.4-fold, respectively in 3D culture conditions compared to 2D culture conditions, whereas *EPCAM* showed the opposite trend (Figure 2B). Additionally, the expression of *CYP3A29* and *CYP1A2*, which are synthesis genes for major CYP enzymes involved in drug degradation in the liver, increased in the 3D cultures by 58.8-fold for *CYP3A29* and in the 2D cultures by 10.9-fold for *CYP1A2*(Figure 2C). Compared to PH, used as a positive control, the 3D organoid cultures of hepatocytes exhibited a smaller decrease in *CYP3A29* expression levels than the 2D cultures.

### 2.3. Physiological Characterization of the Liver using HOs

Immunofluorescence staining highlighted the presence of key liver proteins within the HOs on day 20. Specifically, Alb, a major liver protein labeled with Texas red, was prominently present in the cytoplasm (Figure 3Ai). E-cadherin 1, a hepatocyte-specific intercellular adhesion molecule labeled with fluorescein isothiocyanate, exhibited robust expression along the plasma membrane, allowing us to identify hepatocytes with bipolar nuclei based on this distinctive pattern (Figure 3Aii). Additionally, cytokeratin 19, an intermediate filament protein labeled with AlexFluor 555, displayed expression around the nucleus and throughout some of the cells composing the HOs (Figure 3Aiii).

Various assays were conducted to assess the metabolic, storage, and detoxification capacities of the liver in Hos (Figure 3B). Oil Red O (Figure 3Bi) and PAS (Figure 3Bii) staining confirmed the accumulation of fat and glycogen, respectively, whereas cholesterol metabolism was confirmed using the 1,1’-Dioctadecyl-3,3,3’,3’-Tetramethylindocarbocyanine Perchlorate (Dil)-Ac-LDL assay, which showed a red signal (Figure 3Biii). The xenobiotic detoxification capacity of the HOs was determined via ICG uptake (green) and release (Figure 3Biv).

The preceding results confirm that HOs express key liver proteins and exhibit metabolic, storage, and detoxification functions similar to the liver.

### 2.4. Co-Culture Effect of A-MSCs in Hepatocyte Cultures

A-MSCs were derived from GFP-expressing porcine adipose tissues (Appendix A). Their differentiation into three types of mesodermal cells, namely adipocytes, osteocytes, and chondrocytes, was confirmed using tissue-specific staining and the expression of tissue-specific transcription factors (Appendix A).

An additional 4000 A-MSCs were co-cultured with 20,000 hepatocytes for a 14 d period (Figure 4A). Notably, the group co-cultured with A-MSCs exhibited a significant increase in the number of HOs between days 5 and 7 compared to the HOs cultured alone. Moreover, the diameters of HOs in the A-MSC co-culture group consistently surpassed those of HOs cultured alone from day 7 onwards (Figure 4B). Although the expression of *ALB* and *KRT19*, which are involved in the synthesis of functional and structural liver proteins, remained similar between the two HO groups, cadherin 1 showed a slight decrease in the A-MSC co-culture group compared to the HOs cultured alone. It is worth noting that all proteins associated with these three genes were well expressed in HOs, regardless of whether they were co-cultured with A-MSCs (Appendix A).

### 2.5. Pharmacokinetics and Metabolic Capacity in HOs Co-Cultured with A-MSCs

The activities of the enzymes involved in xenobiotic detoxification were evaluated after treatment with a carcinogen and *CYP1A2* inducer 3-MCT and an antibiotic and *CYP3A29* inducer RIF (Figure 4D). *CYP1A2* expression was 19.4-fold higher in 3-MCT-treated HOs alone than in naïve HOs alone, and 15-fold higher in 3-MCT-treated HOs co-cultured with A-MSCs than in naïve HOs co-cultured with A-MSCs. *CYP3A29* expression was 5.3-fold higher in RIF-treated HOs alone than in naïve HOs alone and 7.3-fold higher in RIF-treated HOs co-cultured with A-MSCs than in naïve HOs co-cultured with A-MSCs.

The metabolic capacity; accumulation of fat, glycogen, and triglycerides; and ability to absorb and degrade ICG (an external drug) were evaluated in HOs co-cultured with A-MSCs (Appendix A). Interestingly, there were no significant differences in these abilities between the two HO groups, regardless of whether they had been co-cultured with A-MSCs. In summary, it was unequivocally confirmed that co-culturing A-MSCs and hepatocytes significantly promoted growth and function in HOs.

### 2.6. Confirmation of In Vitro Expansion Ability in HOs

To test the in vitro expansion ability of the HOs, they were subcultured at a 1:2 ratio every 14 d. Fresh A-MSCs were co-cultured with the HOs at each subculture to compare the results to those of control HOs cultured alone. HOs at passage four (70 d) showed a significant decrease in their ability to form, with almost no formation observed. In contrast, HOs co-cultured with A-MSCs continued to thrive and could be cultured up to passage eight (126 d), albeit with reduced sizes and a smaller number formed, as depicted in Figure 5A. In parallel, we conducted observations of in vitro cultures of hepatocytes in both the single group and the group co-cultured with A-MSCs under 2D culture conditions (Appendix A). In both groups, cell viability significantly declined, with most of the cells dying by passage two.

Alb and urea secretion, as well as the glucose uptake function of the liver, were confirmed using the supernatants recovered from the 2D and 3D (HOs) cultures of hepatocytes (Figure 5B). By day 28 of the in vitro culture process, high expression levels were maintained regardless of the culture method, including in 2D and 3D cultures, and of whether A-MSCs were co-cultured.

Unlike 2D-cultured hepatocytes, 3D-cultured HOs continued to secrete Alb even after 42 d in in vitro cultures. Notably, HOs co-cultured with A-MSCs showed consistent Alb secretion, even after 70 d in in vitro cultures.

In the case of urea secretion, it was confirmed that a larger amount was continuously secreted in the 3D-cultured HOs compared to that in the 2D cultures. In particular, HOs co-cultured with A-MSCs exhibited significant urea secretion over 70 d in vitro, surpassing that of the HO group alone.

Regarding glucose consumption, the 3D HOs consumed a significantly higher amount than the 2D hepatocytes, especially the 3D HOs co-cultured with A-MSCs (Figure 5Biii). Glucose consumption was the highest at passage zero in the 2D culture and at passage one in the 3D culture. Furthermore, consistent glucose consumption was observed in 3D HOs co-cultured with A-MSCs after passage three. Co-culturing HOs with A-MSCs prolonged the lifespans of HOs in vitro and maintained their liver functions. It has been stated that co-culturing hepatocyte organoids (HOs) with A-MSCs extended their lifespan in vitro while maintaining crucial liver functions including albumin synthesis and secretion, urea production, and glycogen storage.

### 2.7. Analysis of Genes Related to Liver Function Maintenance and Lifespan during In Vitro Long-Term Culturing of HOs

The expression of critical genes involved in major protein synthesis in the liver, *ALB* and *TF*, was analyzed (Figure 6A) from passage zero to passage eight, with passages being performed at 14-day intervals. *ALB* expression in HOs co-cultured with A-MSCs was like that in the control group until passage three. In contrast, the expression of *ALB* in HOs decreased sharply from passage two (downregulated by 0.68-fold) compared to that in the control. In HOs co-cultured with A-MSCs, ALB expression ranged from 0.52-fold (at passage three) to 1.06-fold (at passage six) between passages three and seven. The *TF* expression showed an upward trend, regardless of co-culture with A-MSCs, until passage two. In HOs co-cultured with A-MSCs, the *TF* expression was consistently high (≥26.5-fold upregulation) between passages four to eight. The expressions of *CYP3A29* and *CYP1A2* in HOs alone and HOs co-cultured with A-MSCs were similar up to passage three. In HOs co-cultured with A-MSCs, the expressions of both genes increased during the in vitro culture period.

The expressions of the hepatocyte stem cell factor *EPCAM* and the cell lifespan factors telomerase reverse transcriptase (*TERT*)*, TP53*, and *P21* were analyzed in HOs cultured in vitro for an extended period (Figure 6B). The *EPCAM* expression increased during the culture period. Both HOs alone and HOs co-cultured with A-MSCs, *TERT, TP53*, and *P21* showed similar expression patterns until passage three. In HOs co-cultured with A-MSCs, *TERT* maintained a 1.2–2.1-fold expression from passage four to passage eight. The *TP53* expression remained consistently high (≥12.2-fold) in all passages except passage three in HOs co-cultured with A-MSCs while *P21* showed a modest increase (1-2.4-fold) compared to *TP53*.

All genes except *ALB* showed an upward-sloping pattern as the in vitro culture period increased, with the slopes varying among the genes.

## 3. Discussion

Existing research has focused on establishing functional HOs in pig models that can be used directly in experiments. To extend their lifespan and enhance their robustness, we aimed to increase the formation rate and maturation of HOs by co-culturing them with A-MSCs.

Mice are commonly used to study human diseases; however, their body sizes and lifespans are different from those of humans, which limits their ability to mimic human diseases. In contrast, pigs are more similar to humans in body and organ size, making them suitable models for studying human diseases [17,18]. Although 3D primary HO culture systems have been established for both humans and mice, a porcine model is yet to be developed. To the best of our knowledge, our paper is the first to report the development of 3D-cultured pig HOs using Matrigel dome culture conditions and not 3D hepatocyte aggregates [19] or spheroids [17,20].

Previous studies mainly focused on growing cholangiocyte-derived organoids over long periods [21]. However, these attempts did not result in finding organoids capable of fully replicating mature hepatocytes. However, we successfully generated functional mature HOs by optimizing the cell seeding density of pig hepatocytes (20,000 hepatocytes/50 µL Matrigel dome) based on the criteria reported in humans and mice [10]. We found that the PH seeding density and the number of HOs were positively correlated, but also promoted cell death owing to the limited space of the Matrigel dome. HOs closely resembled the liver, as evidenced by the robust upregulation of genes associated with *ALB* synthesis and a major drug-degrading enzyme, *CYP3A29*. Additionally, our HOs demonstrated enhanced liver regeneration capabilities, as indicated by the upregulation of *AFP*, a key fetal liver protein known to be increased during liver damage and regeneration and in human HOs [10].

Interestingly, although we generated HOs using hepatocytes, a cholangiocyte factor protein, cytokeratin 19, was detected in the HOs. This is likely due to the plasticity of hepatocytes, which can transform into bile duct epithelial cells depending on the environment [10,22]. Another biliary epithelial cell factor gene, *EPCAM*, is not only a pluripotent stem cell (PSCs) marker that plays an important role in proliferation and maintenance [23,24,25] but also a hepatic progenitor/stem cell marker [25]. Therefore, the increase in *EPCAM* gene expression in our HOs indicated an improved proliferative capacity in vitro.

Compared to cholangiocyte organoids, HOs are known to exhibit limited in vitro expansion capacity in both murine and human models [10]. Previous reports suggested that the proliferative capacity of HOs derived from adult animals decreases after 2–3 months, possibly because of the relatively short telomere length of the constituent hepatocytes [10,26]. Our findings are consistent with these observations, demonstrating a decline in the proliferative capacity of porcine HOs after 2 months of culture. In HOs from passages zero to three, the expression level of *TERT* ranged from 0.81 to 1.51 times that in PHs. Therefore, we introduced a co-culture system to augment both liver function and the proliferation capacity of HOs. This approach is based on the understanding that hepatocytes can boost their in vitro proliferative capacity in response to injury caused by infection and co-culture with other cells [14,20,27,28]. Our goal was to enhance the efficiency in a favorable direction.

Previous studies have shown that co-culturing hepatocytes with various types of MSCs, including those derived from the bone marrow and umbilical cord, as well as human umbilical vein endothelial cells, can enhance liver function within 3D-culture systems such as spheroids and cell-scaffold structures. However, the continued maintenance of this improved function during long-term in vitro culture remains a challenge, as demonstrated in previous studies [14,20,28]. To the best of our knowledge, our work represents a pioneering advancement in the development of HOs using 3D Matrigel dome cultures in a porcine model. Remarkably, we achieved significant and persistent improvements in liver function by co-cultivating these HOs with GFP-expressing A-MSCs. MSCs secrete crucial soluble factors, including cytokines and extracellular vesicles, that contribute significantly to hepatocyte proliferation and differentiation [15]. Moreover, they play a vital role in enhancing cell–cell adhesion and extra-cellular matrix synthesis, which are both fundamental for organoid formation [14,28]. Therefore, GFP A-MSCs have a positive effect on the in vitro culturing of HOs.

The co-culturing of HOs with A-MSCs extended the proliferation period and maintenance function two-fold compared with using HOs cultured alone. Nevertheless, after ~100 d, clear signs of cell death emerged, marked by an increase in *TP53-P21* and a decrease in *EPCAM* expression, unequivocally signifying a decline in proliferation. *TP53-P21* is a cell death pathway initiated by reactive oxygen species production [29]. In addition, it became evident that liver function diminished after ~100 d in the co-cultures of HOs, as reflected by reduced levels of *ALB, TF*, and *CYP3A29*, which play pivotal roles in major liver protein synthesis and drug metabolism.

Consequently, we successfully established the optimal culture conditions for generating functional HOs applicable to pig models. Our 3D organoid culture system significantly prolonged both the proliferation and maintenance periods of hepatocytes, reaching approximately 100 d, a 3.5-fold increase compared to the conventional 2D culture, which typically lasts only 28 d in vitro. Co-culturing these organoids with A-MSCs within the 3D setup not only boosted proliferation but also doubled the maintenance duration compared to using HOs cultured alone. Moreover, we anticipate that the somewhat limited proliferative capacity of these HOs compared to that of organoids derived from biliary epithelial cells can be offset by cryopreservation. Our observations indicate that these organoids remain viable even after undergoing a one-year freezing process owing to their preserved regenerative ability. This holds great promise for advancing research on human metabolism and establishing scalable cell sources for device development. Additionally, our research will play a vital role in reducing the use of experimental animals for in vitro toxicity assessments of environmental toxins in livestock feed and animal drug development.

## 4. Materials and Methods

### 4.1. Chemicals and Medium

All the chemicals and media used in this study were purchased from Sigma-Aldrich (St. Louis, MO, USA) and Thermo Fisher Scientific (Waltham, MA, USA). Exceptions are indicated unless otherwise specified.

### 4.2. Experimental Samples and Ethics Statements

Hepatocyte-rich cells from a one-month-old female mini-pig were harvested using a modified two-step collagenase perfusion method [30,31] using Liver Perfusion Medium and Liver Digestion Medium. The cells were centrifuged in 90% Percoll solution (Sigma-Aldrich) at 200× *g* and 4 °C to increase the purity of hepatocytes [32]. Finally, they were washed twice with Willam’s medium, and their survivability was assessed using the 0.4% trypan blue exclusion test (Invitrogen). Hepatocytes were cryopreserved at a concentration of 2 × 10^7^/mL using CRYO-GOLD (Revive Organtech, Inc.; Irvine, CA, USA) according to the manufacturer’s protocol and stored in liquid nitrogen until use. All experiments were approved by the Institutional Animal Care and Use Committee of the National Institute of Animal Science (approval number: NIAS20212195) of the Rural Development Administration (RDA), Republic of Korea.

### 4.3. Generation of Hepatocyte Organoids

Frozen hepatocytes were thawed in 37 °C water, then centrifuged to 50× *g* for 5 min and centrifuged twice with Advanced Dulbecco’s modified eagle’s medium (A-DMEM) with 10% fetal bovine serum (FBS) to 50× *g* for 5 min. Cell pellet were suspended in a cold 36% Percoll solution and centrifuged at 150× *g* for 7 min at 4 °C to recover healthy cells. The recovered cell pellet was washed twice with the hepatocyte organoid culture medium, total cell number was calculated, and survivability as evaluated using 0.4% trypan blue exclude test was performed using Luna™ Automated cell counter (Logos Biosystems, Anyang, Gyeonggi, Republic of Korea, L10001). The experiment was performed in three repetitions. All subsequent experiments were conducted using 36% Percoll-treated hepatocytes unless otherwise specified.

Based on the conditions used to produce liver organoids in humans and mice [10], hepatocytes were serially seeded at 1000, 10,000, 20,000, or 50,000 per 50 uL Matrigel matrix (Corning^®^ 354262, Glendale, AZ, USA) to determine the optimal cell seeding concentration. They were cultured with 500 ul of HepatiCult™ Organoid Growth Medium (OGM) (STEMCELL Technologies, Vancouver, BC, Canada) per well (in a 24-well plate) in humidified 5% CO_2_ atmosphere at 37 °C until day 20. The OGM medium was replaced with fresh culture medium every 3 d. This culture method is denoted as the 3D Matrigel dome culture method.

As a criterion for quantification, HOs with diameters of 50 μm or more were observed under a microscope, and the diameters and number of HOs were quantified by repeating this five times on days 5, 7, 14, and 20. Total ribonucleic acid (RNA) was extracted from the two groups seeded with 20,000 or 50,000 hepatocytes for real-time polymerase chain reaction (PCR) experiments on day 20.

### 4.4. D and 3D Organoid Cultures of Hepatocytes for Analysis of Liver-Specific Genes

A total of 20,000 hepatocytes were cultured under two different culture conditions using OGM medium for 14 d: 2D monolayer culture or 3D Matrigel dome culture in 24-well plates. Ear fibroblasts (EFs) from the same individual served as negative controls whereas liver tissue from the same individual and primary hepatocytes, recovered after thawing and being treated with 36% Percoll, were used as positive controls.

### 4.5. Immunofluorescence Staining

HOs cultured in vitro for 14 d were harvested, fixed with ice-cold 3.7% formalin (Sigma-Aldrich) and washed twice with Dulbecco’s phosphate buffered saline (DPBS) without calcium and magnesium (Invitrogen) containing 1% bovine serum albumin (BSA) (Sigma-Aldrich), 0.1% Triton X-100 (Bio-Rad Laboratories Inc., Mississauga, ON, Canada), and 0.05% Tween 20 (Sigma-Aldrich) for 10 min at room temperature (RT) and treated to prevent non-specific binding of antibodies with 3% BSA for 1 h at 37 °C. As shown in Appendix A, samples were treated with each specific primary antibody for albumin, E-cadherin 1, and cytokeratin 19 and kept at 4 °C overnight, then followed the treatment procedure involving appropriate secondary antibodies for 1 h at 37 °C in the dark. For nuclear staining, HOs were treated with 1 µg/mL 4′, 6-diamidino-2-phenylindole (DAPI) (Invitrogen) for 30 min at RT. Finally, cells were mounted with VECTASHIELD^®^ Antifade Mounting Medium (VECTOR, Burlingame, CA, USA, H-1000) and observed under a laser-scanning confocal microscope (Leica, Wetzlar, Germany) or a fluorescence microscope (Leica DMI 6000B, Leica, Wetzlar, Germany).

### 4.6. Liver Function Assay

HOs were fixed in 4% formalin and rinsed twice with Ca^++^Mg^++^-free DPBS. Subsequently, Oil Red O staining (Sigma-Aldrich) was performed to detect neutral triglycerides and lipids while Periodic Acid Schiff (PAS) staining was conducted to assess the glycogen storage ability in the liver tissues. Additionally, fresh HOs underwent an indocyanine green (ICG) clearing test and acetylated low-density lipoprotein (Ac-LDL) uptake assay. These staining techniques and assays were chosen based on a previous study conducted by Ullah et al. [6] and the protocols provided by the manufacturers.

### 4.7. Co-Culturing of HOs with A-MSCs

A-MSCs were isolated from the adipose tissues of transgenic miniature pigs expressing green fluorescent protein (GFP) sourced from Optipharm (Cheongju, Republic of Korea) following the isolation protocol and mesenchymal stem cell confirmation method outlined in a prior study [16].

To assess the impact of mesenchymal stem cell co-culturing on HO generation, hepatocytes and A-MSCs were combined in 50 µL of Matrigel at a 5:1 ratio (hepatocytes:A-MSCs, 20,000:4000). HO generation efficiency was quantified on days 5, 7, 10, and 14, as previously described. Subsequently, HOs were subcultured at a 1:2 ratio every 14 d to evaluate their proliferative potential. At each passage, HOs were harvested for the analysis of liver function-related genes, and supernatants from HOs were preserved at −70 °C for later examination. This experiment was conducted in triplicates. In addition, 2D hepatocyte cultures were developed in parallel under the same conditions for comparison. They were subcultured at a 1:2 ratio every 14 d and kept as the supernatants in a −70 °C deep freezer for subsequent analysis. This experiment was performed in triplicates.

### 4.8. Assessment of Cytochrome P450 (CYP) Activity Using Inducers

Regardless of co-culture, HOs developed on day 13 were treated with OGM supplemented with 10 μM 3-methylcholanthrene (3-MCT), a carcinogen and inducer of CYP family 1 subfamily A (CYP1A) enzyme [33,34], and 25 μM rifampicin (RIF) [5,35], an antibiotic and inducer of CYP family 3 subfamily A member 29 (*CYP3A29*), at 37 °C in a 5% CO_2_ incubator for 72 h [6]. After pooling samples from three repeated experiments, total RNA was extracted from the HOs. The EFs were used as internal controls.

### 4.9. Real-Time Reverse-Transcription Quantitative PCR (RT-qPCR)

Total RNA extraction, copy deoxyribonucleic acid (cDNA) synthesis, and RT-qPCRs were performed according to a modified version of a previously described method [6]. cDNA was synthesized from purified total RNA (40–500 ng). Specific primer sets for various genes were designed and are listed in Appendix A. The samples were analyzed using the *ΔΔCT* method, with hypoxanthine phosphoribosyltransferase 1 serving as the endogenous control. The PCR data are presented as relative quantitation (RQ) values, and the error bars denote RQs ± minimum (min) and RQs ± maximum (max).

### 4.10. Verification of Secretory Ability in Hepatocyte Organoids

Albumin and urea secretion, as well as glucose consumption, were assessed using supernatants from HOs. An Albumin Pig Enzyme-Linked Immunosorbent Assay Kit (ab108794, Abcam, Cambridge, UK) and Urea Assay Kit (ab83362) were used for albumin and urea measurements, respectively, following the manufacturer’s instructions. Glucose consumption was determined by comparing the glucose concentration in fresh hepatic OGM with that in the recovered supernatants. This analysis was conducted using a biochemical analyzer (Catalyst Dx Chemistry Analyzer; IDEXX Laboratories Inc., Westbrook, ME, USA). All experiments were performed in triplicate.

### 4.11. Statistical Analyses

Statistical analyses were performed using Statistical Product and Service Solutions (SPSS) software (The International Business Machines Corporation, Armonk, NY, USA) (ver. 25). Student’s *t*-test for two independent groups and one-way Analysis of Variance (ANOVA) with post-hoc Tukey’s honestly significant difference (HSD) test or Fisher’s least significant difference test for three or more groups were used. All experiments were performed with five replicates. Data on PCRs were expressed as RQ values, and error bars represented RQs ± min and RQs ± max. The significance level was set at *p* < 0.05 statistically. Other data were represented as means ± standard deviation (SD) or standard errors of means (SEMs).

## Figures and Tables

**Figure 1 ijms-24-15429-f001:**
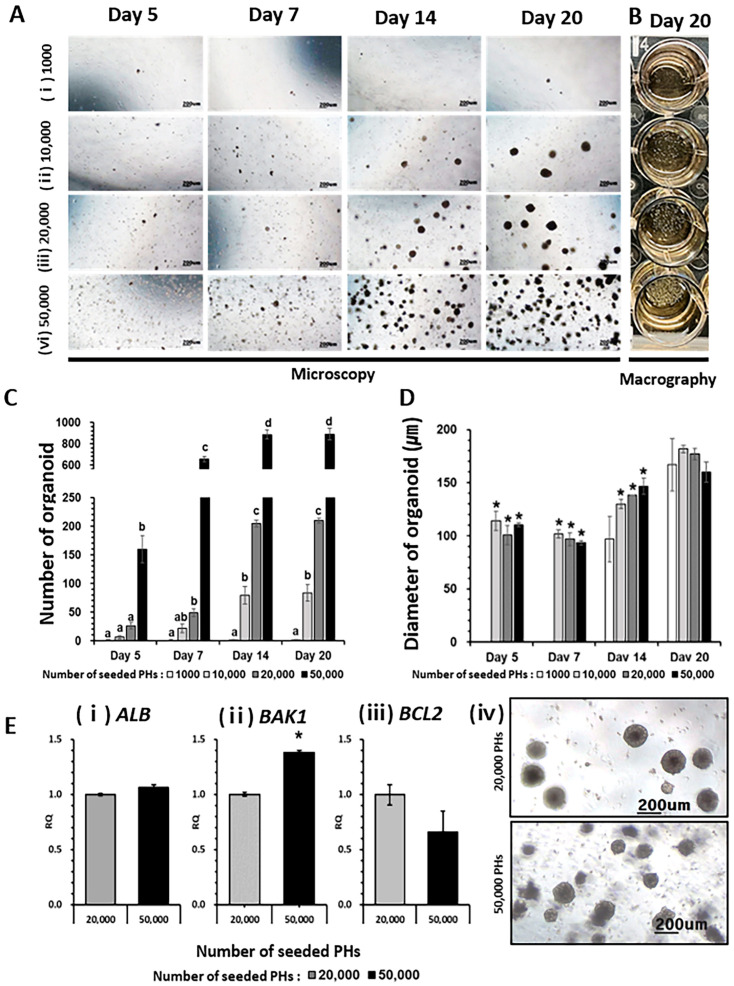
Confirmation of primary hepatocyte (PH) seeding density-dependent hepatocyte organoid (HO) formation. (**A**) PHs were seeded at 1000 (**i**), 10,000 (**ii**), 20,000 (**iii**), and 50,000 (**iv**) cells per 50 µL Matrigel and cultured in vitro for 20 d. They were observed under a phase-contrast microscope. (**B**) Macrography showed HOs observed with the naked eye on day 20. Scale bars represent 200 µm (*n* = 5). (**C**) and (**D**) show the number of HOs and the diameters of the HOs on day 5, 7, 14, and 20, respectively. HOs with a diameter of at least 50 µm were analyzed, and statistical analyses were performed among different cell concentration groups on the same days. Data are presented as means ± standard deviation (SD). Different letters indicate significant differences among groups based on one-way ANOVA followed by Tukey’s HSD multiple range tests (SPSS 25, *p* < 0.05, *n* = 5), and asterisks (*) above each bar show statistically significant differences between groups at *p* < 0.05 (*n* = 5). (**E**) Expression of *ALB* (**i**) and apoptosis-related genes (pro-apoptosis: *BAK1* (**ii**), anti-apoptosis: *BCL2* (**iii**)) in groups (**iv**) of 20,000 and 50,000 seeded hepatocytes. Error bars represent the minimum (min) and maximum (max) relative quantification (RQ) levels around the mean RQ expression levels. Statistical analysis was performed using Student’s *t*-test on SPSS 25 (*n* = 5), and significant differences at *p* < 0.05 are denoted by asterisks (*) above the bars.

**Figure 2 ijms-24-15429-f002:**
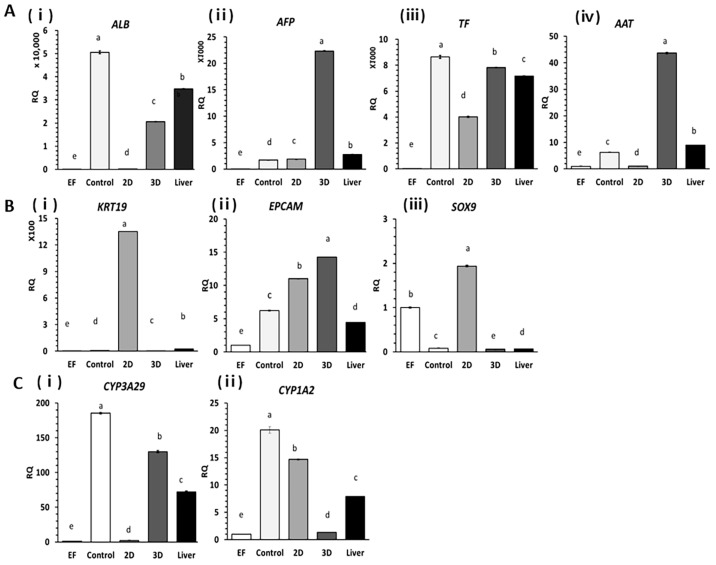
Gene expression profiling of major liver genes based on primary hepatocyte (PH) culture conditions. PHs (2 × 10^4^ cells) were cultured under 2D or 3D culture conditions for 14 d in the same culture media. The culture medium was replaced with fresh medium every three days. Genes involved in the synthesis of major proteins synthesized in the liver, such as *ALB* (**i**), *AFP* (**ii**), *TF* (**iii**), and *AAT* (**iv**), were analyzed (**A**). Cholangiocyte-specific genes *KRT19* (**i**), *EPCAM* (**ii**), and *SOX9* (**iii**) were analyzed (**B**). Genes involved in the synthesis of key enzymes involved in external drug degradation, such as *CYP3A29* (**i**) and *CYP1A2* (**ii**), were analyzed (**C**). Error bars represent minimum (min) and maximum (max) relative quantification (RQ) levels around mean RQ expression levels. Different letters indicate significant differences among groups based on one-way ANOVA followed by Tukey’s HSD multiple range test (SPSS 25, *p* < 0.05, *n* = 5). The open, light-gray, medium-gray, dark-gray, and black bars represent ear fibroblasts, PHs, hepatocytes cultured in 2D cultures, HOs cultured in 3D cultures, and liver tissue, respectively.

**Figure 3 ijms-24-15429-f003:**
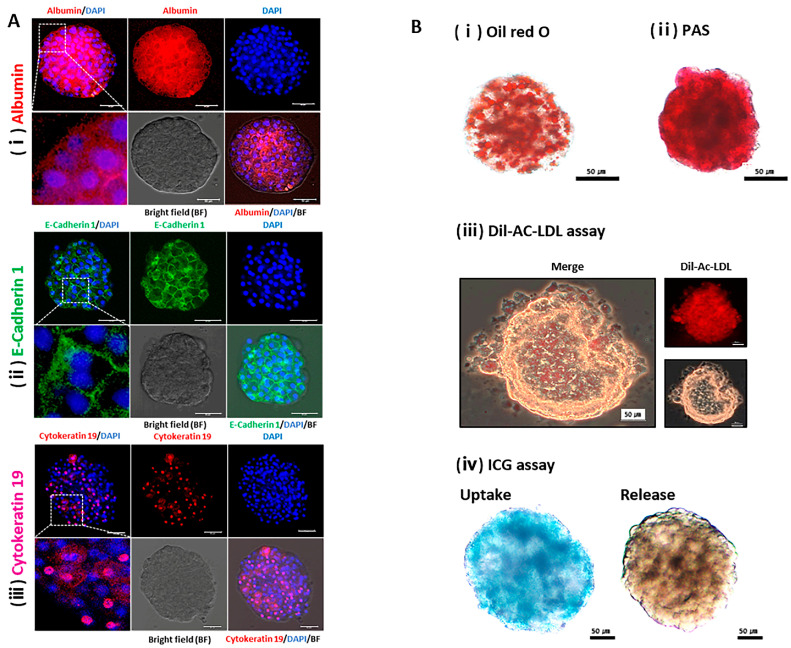
Analysis of the functional and structural proteins and physiological functions of the liver in hepatocyte organoids. (**A**) Representative immunofluorescence image of (**i**) albumin (purple), (**ii**) E-cadherin 1 (green), and (**iii**) cytokeratin 19 (purple) on day 20. Nuclei were counterstained with DAPI. Scale bars represent 50 µm. BF: bright field. The white dashed boxes were magnified in the lower panel. (**B**) Physiological functions of hepatocyte organoids: (**i**) Oil Red O staining was performed to detect lipid droplets (red). (**ii**) Glycogen accumulation (red) was evaluated via Periodic Acid Schiff (PAS) staining. (**iii**) Low-density lipoproteins (LDLs, red) were detected via Dil-acetylated LDL (Dil-Ac-LDL) assay. (**iv**) Uptake (blue) and release capacity of indocyanine green (ICG) were evaluated for ICG clearance assay. Each scale bar represents 50 µm.

**Figure 4 ijms-24-15429-f004:**
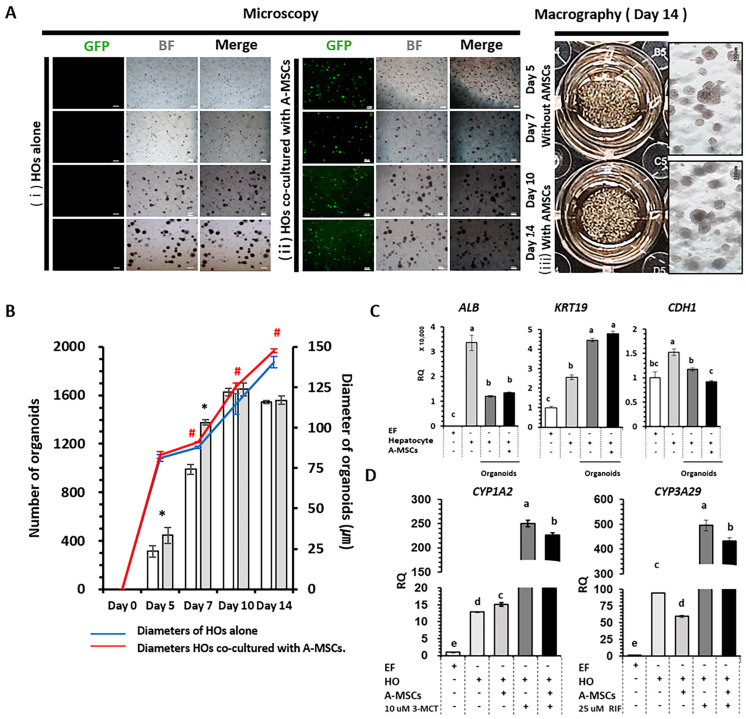
The effects of co-cultured adipose tissue-derived mesenchymal stem cells (A-MSCs) on hepatocyte organoid (HO) generation. (**A**) HOs were formed by co-culturing 4000 GFP-expressing AMSCs with 20,000 hepatocytes in a 1:5 ratio for 14 d. (**Ai**) HOs alone; (**Aii**) HOs co-cultured with A-MSCs; (**Aiii**) macroscopic observation of HOs cultured for 14 d. GFP, BF, and Merge are abbreviations for green fluorescent protein, bright field, and the combined image of the two. Scale bars represent 200 µm. (**B**) HO numbers and diameters at 5, 7, 10, and 14 d: open/gray bars for the number of HOs alone/co-cultured with A-MSCs; red/blue lines for the diameters of HOs alone/co-cultured with A-MSCs. Data displayed are means ± standard deviation (SD). *p* < 0.05 (*n* = 5). Asterisks (*) and hashes (#) above bars and lines indicate statistical significance at *p* < 0.05. (**C**) Expression of *ALB, KRT19*, and *CDH1* in HOs. (**D**) Evaluation of *CYP1A2* and *CYP3A29* drug metabolism after treatment with 10 µM 3-methylcholanthrene (3-MCT) and 25 µM rifampicin (RIF) for 3 d. After pooling samples from three repeated experiments, five mechanical repetitions were performed. Error bars represent the minimum (min) and maximum (max) relative quantification (RQ) levels around the mean RQ expression levels. Different letters denote significant differences among groups (*p* < 0.05). The minus (−) and plus (+) signs indicate the absence and presence of cells or drugs, respectively.

**Figure 5 ijms-24-15429-f005:**
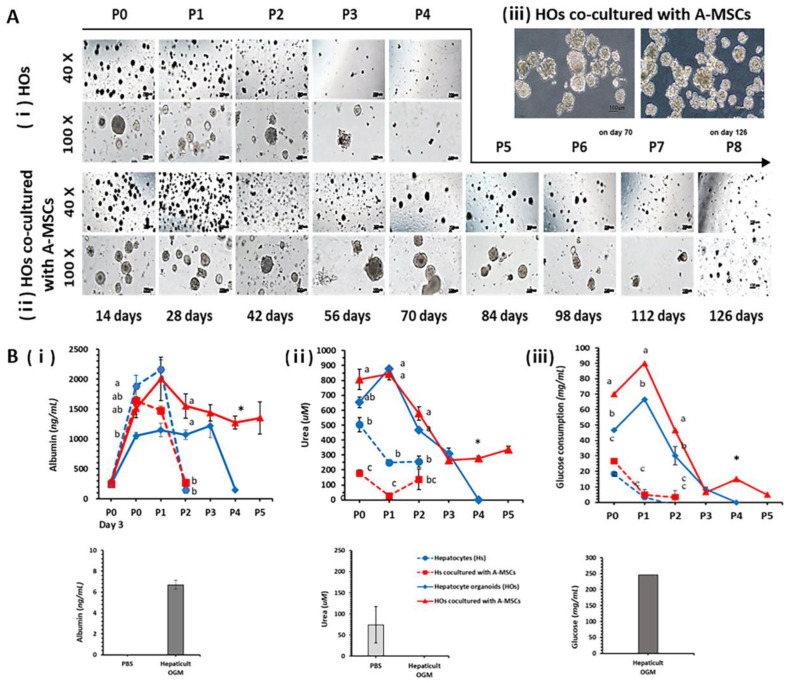
Evaluation of the in vitro proliferation and secretory abilities of hepatocyte organoids (HOs) co-cultured with A-MSCs. (**A**) HOs were subcultured at a 1:2 ratio every 14 d. At each subculture step, fresh A-MSCs (**ii**) were co-cultured to be compared with the control HOs alone (**i**). (**iii**) HOs co-cultured with A-MSCs after removal of Matrigel on day 70 and day 126. (**B**) Every 14 d, the culture medium supernatant was collected, and the levels of albumin (**i**), urea (**ii**), and residual glucose (**iii**) were analyzed using a biochemical analyzer. In addition, 2D cultures of PHs were prepared in parallel under the same conditions for comparison. The cells were subcultured at a 1:2 ratio every 14 d. For albumin, the results from day 3 were added when the cells were fully attached to the plate. Glucose consumption was determined by measuring the glucose concentration in the fresh HepatiCult™ Organoid Growth Medium (Hepaticult OGM) and subtracting the glucose concentration measured in the recovered supernatant. Blue and red dotted lines indicate primary hepatocytes (PHs) cultured alone or co-cultured with A-MSCs in 2D cultures, respectively. Conversely, the blue or red solid lines represent HOs cultured alone or co-cultured with A-MSCs in 3D cultures. Light- and dark-grey bars correspond to PBS and Hepaticult OGM, respectively. Different letters denote statistical significance among groups whose data were collected on the same date (*p* < 0.05; *n* = 3). Asterisks (*) indicate significant differences between the 3D organoid groups (*p* < 0.05; *n* = 3). Data are presented as means ± standard errors of the means (SEMs).

**Figure 6 ijms-24-15429-f006:**
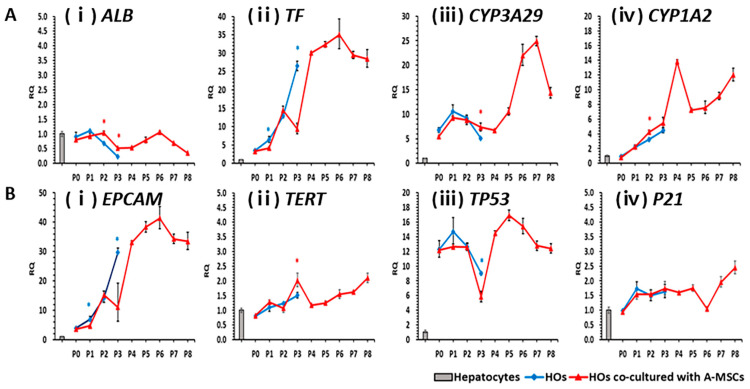
Gene profiling based on liver metabolism, lifespan, and apoptosis in proliferative hepatocyte organoids (HOs) over varying in vitro culture periods. HOs were subcultured every 14 d at a 1:2 ratio, and fresh A-MSCs were added during each subculture step for comparison with the control HOs alone. Gene expression analysis was conducted on the harvested organoids before they were subcultured at each passage. Gray bars represent PHs while blue and red lines represent HOs cultured alone or co-cultured with A-MSCs, respectively. A and B represent the expression levels of *ALB* (**Ai**), *TF* (**Aii**), *CYP3A29* (**Aiii**), and *CYP1A2* (**Aiv**). B represents the expression levels of *EPCAM* (**Bi**), *TERT* (**Bii**), *TP53* (**Biii**), and *P21* (**Biv**), as determined using real-time quantitative PCRs. Error bars represent minimum (min) and maximum (max) relative quantification (RQ) levels around mean RQ expression levels. Asterisks (*) indicate significant differences between groups (*p* < 0.05, *n* = 5).

## Data Availability

All data generated or analyzed during this study are included in this published article and its Appendix A.

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
