# Peer review of "Adipose Tissue-Derived Mesenchymal Stem Cells Extend the Lifespan and Enhance Liver Function in Hepatocyte Organoids"

_ijms, 2023, doi:10.3390/ijms242015429_

Round 1

Reviewer 1 Report

This work seems interesting in some aspects and distracting in others. It is necessary to pay more attention to the presentation of the results in order to make the results more meaningful.

- In paragraph 2.2 it is not clear why CYP genes increase from 2D to 3D culture. Is it physiological or induced? This should be explained in more detail.

- Paragraphs 2.3 and 2.4 lack a concluding sentence explaining and justifying the results obtained.

- All results seem to be better explained in the caption under the figure than in the corresponding paragraph. This should be improved.

Reviewer 2 Report

General remarks:

Overall, the work appears to have been well conducted: the reported experiments have been appropriately conceived, performed, and interpreted. The conducted assays, including immunofluorescence, gene expression analysis, staining, and growth analysis are what you would expect from a work involving the characterization of hepatocyte organoids. Thus, I do not have any major remarks from the technical point of view. However, this work appears to be mostly incremental, with very few elements of novelty. In fact, it is well known that co-culture with mesenchymal stem cells has a positive effect on the health/growth/regeneration of nearly any tissue, so the findings of this work are not surprising at all. Moreover, the benefit of organoids is that they can be relatively easily obtained from human patients and allow the replacement of animal models with organ models that reflect human physiology, albeit in a simplified setting. It appears to me that it is rather pointless seeking to reduce the number of pigs used in research with pig organoids, rather than with human organoids. Therefore, it is the editor's call to determine whether the manuscript has a sufficient novelty factor to make it suitable for publication. If this is the case, the manuscript reports in my opinion an appropriately conducted work, again with very few elements of novelty.

Minor remarks:

In Figure 1A and in Figure 4A the microscopy images of the seeded hepatocyte are too small and with excessive contrast, so the details of the cells are not visible. The authors should present representative regions of the plate at higher magnification and using a proper phase contrast imaging.

The English is understandable and no major editing of the manuscript is required.
